# Landauer Principle and General Relativity

**DOI:** 10.3390/e22030340

**Published:** 2020-03-16

**Authors:** Luis Herrera

**Affiliations:** Instituto Universitario de Física Fundamental y Matematicas, Universidad de Salamanca, 37007 Salamanca, Spain; lherrera@usal.es

**Keywords:** Landauer principle, information theory, general relativity, gravitational radiation

## Abstract

We endeavour to illustrate the physical relevance of the Landauer principle applying it to different important issues concerning the theory of gravitation. We shall first analyze, in the context of general relativity, the consequences derived from the fact, implied by Landauer principle, that information has mass. Next, we shall analyze the role played by the Landauer principle in order to understand why different congruences of observers provide very different physical descriptions of the same space-time. Finally, we shall apply the Landauer principle to the problem of gravitational radiation. We shall see that the fact that gravitational radiation is an irreversible process entailing dissipation, is a straightforward consequence of the Landauer principle and of the fact that gravitational radiation conveys information. An expression measuring the part of radiated energy that corresponds to the radiated information and an expression defining the total number of bits erased in that process, shall be obtained, as well as an explicit expression linking the latter to the Bondi news function.

## 1. Introduction

The Landauer principle [1] is one of the cornerstone of the modern theory of information. It states that the erasure of information is always accompanied by dissipation of energy. More specifically, according to this principle the erasure of a bit of information stored in a given system, entails the dissipation of a minimal amount of energy *E* given by,
(1)E=kTln2,
where *k* denotes the Boltzman constant and *T* is the temperature of the environment. By erasure we means the reseting of the system to a predetermined state. The ensuing change in the entropy produces energy dissipation. The total amount the energy dissipated in this process may depend on the erasure procedure, however the relevant point is that its value cannot be lower that (Equation 1).

Before proceeding farther, it is worth mentioning that some authors refer to this principle as the Brillouin principle (see for example Reference [2]) arguing that the content of the principle was first put forward by Brillouin in Reference [3]. Here, without entering into the historical roots of this issue, we shall follow the notation adopted by the majority of researchers, and will refer to it as the Landauer principle.

Although the relevance of this principle has been questioned in the past, based on the argument that it is not independent of the second law of thermodynamics (see Reference [4] and references therein), the fact is that it not only allows an “informational” reformulation of thermodynamics, as stressed in Reference [4], but also establishes a deep link between information theory and different branches of science [5,6,7]. Besides it has been claimed that a derivation of Landauer’s principle without direct reference to the second law of thermodynamics, is possible [8]. However, whether the Landauer’s principle is just another version of the second law of thermodynamics or is a fundamental principle on its own, is not of our concern in this work. The fact that this principle allows to address different physical problems from the point of view of information theory, provides, in our opinion, a sufficiently strong motivation to use it and to delve deeper into its consequences.

In this work we endeavour to apply the Landauer principle in the treatment of different important issues in the study of self-gravitating system. We shall describe the gravitational interaction by means of the Einstein theory (general relativity), however most of the conclusions extracted from our work, could be easily extended to other theories of gravitation.

We shall first consider the consequences derived from the fact that there is a mass associated to a bit of information. We shall analyze this question in the context of the “orthodox” black hole model, as well as in the context of the hyperbolically symmetric black hole.

Next, we shall see how the Landauer principle helps to understand the well known, though intriguing, fact that different congruences of observers, locally related by a Lorentz boost, describe different pictures of the same space-time. In particular, the Landauer principle is crucial to understand why a given system may be isentropic for comoving observers, while the same system is dissipative as seen by “tilted” (non-comoving) ones.

Finally, we shall analyze some aspects of gravitational radiation in the light of the Landauer principle. This will confirm the well known fact that the emission of gravitational radiation is always accompanied by dissipative processes. Also, we shall be able to determine what part of the radiated energy is related to the erasing of information in the process of radiation, and how much information is erased during this process.

## 2. The Mass of a Bit of Information and the Back Hole Picture

The question about the mass of information has been discussed by several authors (see References [9,10] and references therein). However the expression for the mass assigned to a bit of information, deduced from a straightforward application of the Landauer principle was obtained for the first time in Reference [11]. There, it was proposed that a mass given by
(2)$$M_{bit}=\frac{kT}{c^{2}}\ln2,$$
where *c* denotes the speed of light in vacuum, be assigned to a bit of information.

Two comments are in order at this point:The Landauer principle establishes a lower bound of the energy that must be dissipated as a consequence of the erasing of one bit of information. Of course, depending on the erasure procedure, it could be that larger amounts of energy are dissipated during that process. The important fact to keep in mind is that independently on the erasure procedure, at least, the amount of energy given by (Equation 1) must be dissipated. Based on this fact we assign to any bit of information the energy (Equation 1).Once we have assigned an explicit amount of energy to a bit of information, then from the well known fact from relativistic dynamics that a mass $E/c^{2}$ has to be ascribed to an energy *E* (see for example Reference [12] page 113, or Reference [13] page 143), the ensuing associated mass (Equation 2) follows at once.

Later on, the same expression was proposed in Reference [14], where a specific experiment intended to verify the above expression is described. Some interesting consequences derived from (Equation 2) have been discussed in Reference [4]. Here we would like to explore the possible consequences ensuing from the fact that information has mass, regarding the behaviour of the information flow across the horizon in the gravitational field of a spherically symmetric mass distribution. However, before entering into the discussion we need an expression for the Landauer principle in a gravitational field.

If the system is embedded in a gravitational field, then as it was shown in Reference [15], in the weak field approximation, the minimal amount of dissipated energy which according to the Landauer principle is produced during the erasing of a bit of information, should be given by
(3)$$E=kT\left(1+\frac{\phi }{c^{2}}\right)\ln2,$$
where ϕ denotes the Newtonian (negative) gravitational potential. Thus, to take into account a gravitational field, amounts to replace the temperature *T* by the so called Tolman temperature, which in the weak field limit reads $T\left(1+\frac{\phi }{c^{2}}\right)$. This is the quantity required to be constant to ensure thermodynamic equilibrium [16]. The idea behind this concept is based on the fact that according to special relativity, thermal energy should have inertia, and therefore should tend to displace to regions of lower gravitational potential implying that a temperature gradient is necessary in order to prevent a flow of heat from regions of higher to lower gravitational potential. This “inertial” effect of the thermal energy was later on confirmed in the relativistic transport equations proposed in References [17,18,19], and as a matter of fact should appear in any relativistic transport equation.

The expression (Equation 3) can be easily generalized to a gravitational field of an arbitrary intensity, as
(4)$$E=kT\sqrt{|g_{tt}|}\ln2\Rightarrow M_{bit}=\frac{kT\sqrt{|g_{tt}|}}{c^{2}}\ln2,$$
where $g_{tt}$ is the tt component of the metric tensor. Indeed, the Tolman temperature is defined by $T\sqrt{|g_{tt}|}$, which in the weak field limit becomes $T(1+\frac{\phi }{c^{2}})$. The condition of thermodynamic equilibrium (Tolman condition) requiring the gradient of the Tolman temperature to vanish now reads $\vec{\nabla }\left(T\sqrt{|g_{tt}|}\right)=0$.

Let us now consider the gravitational field produced by a spherically symmetric distribution of matter. It is described by a well known solution of the Einstein vacuum equations (the Schwarzschild solution), which in polar coordinates reads
(5)$$\begin{array}{ccc} ds^{2} & = & -\left(1-\frac{2m}{R}\right)dt^{2}+\frac{dR^{2}}{\left(1-\frac{2m}{R}\right)}+R^{2}d\Omega^{2},\\ d\Omega^{2} & = & d\theta^{2}+\sin^{2}\theta d\phi^{2}, \end{array}$$
where the only parameter of the solution *m* may be identified as the total mass (energy) of the source.

For R>2m the above solution is spherically symmetric, that is, it admits the three Killing vectors defining the spherical symmetry given by:(6)$$\begin{array}{c} K_{\left(1\right)}=\partial_{\phi },\;K_{\left(2\right)}=-\cos\phi \partial_{\theta }+\cot\theta \sin\phi \partial_{\phi },\;K_{\left(3\right)}=\sin\phi \partial_{\theta }+\cot\theta \cos\phi \partial_{\phi }. \end{array}$$
and is static, that is, it admits the time-like Killing vector associated to the time invariance defined by
(7)$$K_{\left(0\right)}=\partial_{t}.$$
At R=2m (the horizon) an apparent singularity appears in the metric component $g_{RR}$. However, as is well known this is a non-physical singularity that may be removed by a coordinate transformation, which allows to extend the solution to the whole space-time, beyond the R>2m region. There is though, a price to pay when applying this procedure, which consists in the fact that the space-time in the region within the horizon (R<2m) is necessarily non-static (see References [12,20] for a discussion on this point). In other words, any coordinate transformation removing the above mentioned apparent singularity, breaks the time invariance in the region R<2m [21].

In the context of the standard black hole description above (with vanishing angular momentum), two main conclusions deserve to be stressed:If we assume that a bit of information is endowed with a mass according to (Equation 4), then it is obvious that no information can cross the horizon outwardly.All particles within the horizon are bound to reach the center at some finite proper time, leading to the formation of a true singularity at R=0.

The two conclusions above are in full agreement with the results obtained by Hawking [22,23,24] indicating that the quantum radiation emitted by the black hole is nearly thermal, that is, it conveys no information, since indeed no information can cross the horizon outwardly (always in the context of the standard picture of the Schwarzschild black hole).

However, an alternative picture to the Schwarzschild black hole has been recently put forward in References [25,26]. This proposal considers the whole space-time divided in two regions separated by the horizon. At the outside of it (R>2m), the spacetime is described by the usual Schwarzschild solution (Equation 5), whereas inside it (R<2m) the space-time is described by the solution
(8)$$\begin{array}{ccc} ds^{2} & = & \left(\frac{2m}{R}-1\right)dt^{2}-\frac{dR^{2}}{\left(\frac{2m}{R}-1\right)}-R^{2}d\Omega^{2},\\ d\Omega^{2} & = & d\theta^{2}+\sinh^{2}\theta d\phi^{2}. \end{array}$$

The relevant fact is that the above metric is static (i.e., it admits the time-like Killing vector (Equation 7)), but is not spherically symmetric, instead it is hyperbolically symmetric, meaning that it admits the three Killing vectors,
(9)$$\begin{array}{ccc} K_{\left(1\right)} & = & \partial_{\phi },\;K_{\left(2\right)}=-\cos\phi \partial_{\theta }+\coth\theta \sin\phi \partial_{\phi },\\ K_{\left(3\right)} & = & \sin\phi \partial_{\theta }+\coth\theta \cos\phi \partial_{\phi }. \end{array}$$

Furthermore its signature is −2 implying that at the horizon there is a change of symmetry and signature. The main motivation to propose this alternative picture, is based on the belief that any equilibrium final state of a physical process should be static, and therefore we should expect to have a static solution in the whole space-time, not only in the region R>2m.

A systematic study of the timelike geodesics in the space-time described by (Equation 8) was carried out in Reference [26]. Two main conclusions emerge from this study, in relationship with our discussion here, namely:Now, unlike the standard picture, massive particles can cross the horizon outwardly, though only along the θ=0 axis.Test particles not only are not condemned to displace to the center, more so they cannot reach the center for any finite value of energy.

According to the first point above, we have that a flux of information from the inside of the horizon to the outside of it, is possible which is clearly at variance with the result obtained in the “classical” black hole picture. Furthermore, assuming that the picture above is correct, the dissipated energy predicted by the Landauer principle, associated to the change of information within the horizon, could also flow along the axis θ=0, and such energy flow could be in principle observable.

Thus we have seen that a fundamental issue in the theory of gravity such as the global picture of the Schwarzschild black hole, may be raised in terms of the theory of information, and in particular invoking the Landauer principle. The ensuing observable consequences of such approach might help to elucidate the quandary about the real structure of the Schwarzschild black hole.

## 3. Dissipation and the Information Stored by Different Congruences of Observers

The Einstein theory of gravitation (general relativity) has a manifestly covariant form, that is, the corresponding field equations are written in terms of tensors. This fact implies in its turn that they may be used by (they are valid for) any congruence of observers. Of course, the fact that the field equations of the theory are covariantly written does not mean that the picture of a given physical system, provided by different congruences of observers, would be the same. As a matter of fact, profound differences in the description of some physical systems may appear when these are described by different congruences of observers. Perhaps one of the most striking examples of this situation is provided by the description of fluid sources of some space-times, given by two different congruences of observers related to each other by a Lorentz boost. More specifically, we have in mind the case when one of the congruences corresponds to comoving with the fluid observers, and the other is obtained by a Lorentz boost of the latter, the so called “tilted” congruence. This issue has been extensively discussed in the literature (see References [27,28,29,30,31,32,33,34,35,36,37,38,39,40] and references therein). An important point has to be stressed here: while it is true that comoving and tilted congruences are related by a Lorentz boost, the important point to keep in mind is that the comoving observer is essentially different from the tilted one in that the former assigns a specific value to the three-velocity of any fluid element (0), leading to the obvious fact that the number of degrees of freedom is lesser than in the tilted congruence.

The main idea behind the situation described above resides in the fact that different congruences of observers assign, to any fluid element, different four-velocity vectors in terms of which the energy momentum tensor decomposes. Thus, for any fluid source of a given solution of the Einstein equations, the two energy-momentum tensors associated to the tilted and the comoving congruence respectively, are equal because of the Einstein equations, but are expressed in terms of different physical variables, leading to completely different pictures of the fluid.

Indeed, let us consider a congruence of observers which are comoving with an arbitrary fluid distribution. For this congruence the four-velocity, in some globally defined coordinate system, reads
(10)$$V^{\mu }=\left(V^{0},0,0,0\right),$$
where greeks indices ran from 0 to 3, with 0 corresponding to the time component. In order to obtain the four-velocity corresponding to the tilted congruence (in the same globally defined coordinate system), one has first to perform a (locally defined) coordinate transformation to the Locally Minkowskian Frame (LMF), after that a Lorentz boost is applied to the LMF to obtain the tilted LMF. Finally, we have to perform a transformation from the tilted LMF, back to the (global) frame associated to the line element under consideration. Doing so we obtain the four-velocity corresponding to the tilted congruence in the original globally defined coordinate system (say ${\tilde{V}}^{\alpha }$).

More specifically, the procedure described above, goes as follows. Denoting by $L_{\mu }^{\nu }$ the local coordinate transformation matrix, and by ${\bar{V}}^{\alpha }$ the components of the four velocity in the LMF, we have:(11)$${\bar{V}}^{\mu }=L_{\nu }^{\mu }V^{\nu }.$$

Next, let us perform a Lorentz boost from the LMF associated to ${\bar{V}}^{\alpha }$, to the (tilted) LMF with respect to which a fluid element is moving with some, non-vanishing, three-velocity.

Thus the four-velocity in the tilted LMF is defined by:(12)$${\tilde{\bar{V}}}_{\beta }=\Lambda_{\beta }^{\alpha }{\bar{V}}_{\alpha },$$
where $\Lambda_{\beta }^{\alpha }$ denotes the Lorentz matrix.

Finally, we have to apply to ${\tilde{\bar{V}}}_{\beta }$, a transformation defined by the inverse of $L_{\mu }^{\nu }$, to obtain ${\tilde{V}}^{\alpha }$.

Let us now consider a given spacetime, which according to comoving observers, is sourced by a dissipationless anisotropic fluid distribution, so that the energy momentum-tensor in the “canonical” form reads:(13)$$\begin{array}{ccc} T_{\alpha \beta } & = & \left(\mu +P\right)V_{\alpha }V_{\beta }+Pg_{\alpha \beta }+\Pi_{\alpha \beta }, \end{array}$$
where as usual, $\mu ,P,\Pi_{\alpha \beta },V_{\beta }$ denote the energy density, the isotropic pressure, the anisotropic stress tensor and the four velocity, respectively.

Instead, as it has been shown in References [29,30,37], for the tilted observers the fluid distribution is described by the energy momentum tensor:(14)$${\tilde{T}}_{\alpha \beta }=\left(\tilde{\mu }+\tilde{P}\right){\tilde{V}}_{\alpha }{\tilde{V}}_{\beta }+\tilde{P}g_{\alpha \beta }+{\tilde{\Pi }}_{\alpha \beta }+{\tilde{q}}_{\alpha }{\tilde{V}}_{\beta }+{\tilde{q}}_{\beta }{\tilde{V}}_{\alpha }.$$

The above expression corresponds to an anisotropic fluid, dissipating energy through the heat flux vector ${\tilde{q}}^{\beta }$.

Since both expressions (Equation 13) and (Equation 14), define the source of the same space-time, they must be equal, which provides the relationship between the physical variables measured by the comoving observer and the physical variables measured by the the tilted one. Furthermore, if the space-time is not spherically symmetric, then if the comoving congruence is vorticity-free, it may occur that the corresponding tilted congruence is endowed with vorticity (see References [30,37]). In this latter case it is possible that gravitational radiation could be detected by tilted observers, whereas it is absent in the picture described by comoving ones (see Reference [37]).

Thus we have to cope with the intriguing fact that tilted observers detect dissipation in a system that appears non-dissipative for comoving observers. The interesting point is that information theory, and in particular the Landauer principle, provide the clue to solve this quandary.

Indeed, to explain such difference in the description of a given system, as furnished by different congruences of observers, it has been put forward in Reference [35] that the origin of this strange situation resides in the fact that passing from one of the congruences to the other, we have to take into account the fact that both congruences of observers store different amounts of information. Therefore, when we apply the operation described above, transforming comoving observers, which assign zero value to the three-velocity of any fluid element, into tilted observers, for whom the three-velocity represents another degree of freedom, we have to keep in mind that according to the Landauer principle the erasure of the information stored by comoving observers (vanishing three velocity), when going to the frame of tilted observers, entails dissipation, thereby explaining the presence of dissipative processes (included gravitational radiation) observed by the latter.

An alternative (though equivalent) approach to understand the origin of this strange situation, also based in the Landauer principle, is suggested by the resolution of the well known paradox of the Maxwell’s demon [41].

Roughly speaking the Maxwell’s demon is a small “being” living in a cylinder filled with a gas and divided in two equal portions by a partition with a small door. The demon is entitled to open the door when the molecules come from the right, while closing it when the molecules approach from the left. Doing so, the demon is able to concentrate all the molecules on the left, reducing the entropy by Nkln2 (where *N* is the number of molecules, and *k* is the Boltzman constant), thereby violating the second law of thermodynamics.

Bennet [42] solved the paradox by showing that the irreversible act which prevents the violation of the second law, is the restoration of the measuring apparatus (by means of which the selection of molecules is achieved), to the standard state previous to the state where the demon knows from which side any molecule comes from. In other words, if we consider the whole system (demon + the gas in the cylinder), the information possessed by the demon before selecting the molecules is smaller than the information after this process has been achieved. Thus, in order to return to the initial state of the demon, the acquired information has to be erased, and once again the Landauer principle implies that to get the demon’s mind back to its initial state, energy has to be dissipated.

The argument above to solve the Maxwell’s demon paradox can be used, after a suitable adaptation, to explain the appearance of dissipative processes in the tilted congruence. Thus, the Maxwell’s demon state before knowing where the molecule is coming from, and the tilted observers are equivalent: in both cases a piece of information is missing. Instead, the Maxwell’s demon state after knowing where the molecule is coming from, is equivalent to comoving observers: they both have acquired additional information.

Therefore passing from comoving to tilted observers, as well as returning the demond’s mind to its initial state, requires the erasure of the acquired information, leading to the observed dissipative processes. This unravel the Maxwell’s demon paradox, and also explains why tilted observers detect dissipation, in systems which are isentropic according to comoving observers.

A particularly illuminating example of the above mentioned effect is provided by the analysis of the departure from equilibrium at the shortest time scale at which the first signs of dynamic evolution are detected.

Thus, let us consider a spherically symmetric fluid distribution which is initially in equilibrium. We shall assume that, for a reason which is not relevant for the discussion, at some initial time (say $t_{0}$) the system abandons such a state. At $t_{0}$ the clock starts to run, and we stop it as soon as we detect the first sign of dynamic evolution. The scale time under consideration is defined by the time interval measured by our clock. We shall take a snapshot of the system, just after it has abandoned the equilibrium, where “just after” means on the smallest time scale at which we can detect the first signs of dynamical evolution.

In the theory of dissipative fluids, there exist three fundamental time scales, each of which is endowed with a distinct physical meaning, namely: the hydrostatic time (sometimes also called the hydrodynamic time), the thermal relaxation time and the thermal adjustment time.

The hydrostatic time is the typical time in which a fluid element reacts to a perturbation of hydrostatic equilibrium, it is of the order of magnitude of the time taken by a sound wave to propagate through the whole fluid distribution.

The thermal relaxation time is the time taken by the system to return to the steady state in the heat flux, after it has been removed from it.

Finally, the thermal adjustment time is the time it takes a fluid element to adjust thermally to its surroundings. It is of the order of magnitude of the time required for a significant change in the temperature gradients.

We shall evaluate the system at a time scale which is smaller than the three time scales described above. It should be emphasized that such a time scale is chosen heuristically. Thus, as mentioned before, if no sign of evolution could be detected within this time scale, it should be enlarged until these signs appear. As we shall see below, such signs do appear within the time scale under consideration, for the tilted observer but no for the comoving one.

To prove our point we need a causal relativistic transport equation, here we shall use the one obtained from the Israel-Stewart theory [19], which reads
(15)$$\tau h_{\nu }^{\mu }q_{;\beta }^{\nu }V^{\beta }+q^{\mu }=-\kappa h^{\mu \nu }\left(T_{,\nu }+Ta_{\nu }\right)-\frac{1}{2}\kappa T^{2}{\left(\frac{\tau V^{\alpha }}{\kappa T^{2}}\right)}_{;\alpha }q^{\mu },$$
where τ, κ, *T*, $a_{\nu }$, $h^{\mu \nu }$ denote the relaxation time, the thermal conductivity, the temperature, the four-acceleration and the projector on the plane orthogonal to the four-velocity, respectively.

Let us now consider a spherically symmetric fluid distribution which is forced to abandon the equilibrium and let us take a “snapshot” of the system immediately after that departure, within the indicated time scale, which implies that the Tolman condition is still valid. This is so because of the fact that our time scale is smaller than the relaxation time, and therefore the temperature gradients have the same values they had in equilibrium. Then we obtain for the “tilted” observer
(16)$$\tau \dot{q}=-\kappa T\dot{\omega },$$
where dot denotes the time derivative, ω is the velocity of a fluid element in the tilted LMF, and *q* is the radial component of the heat flux measured in the tilted LMF (see Equation (41) in Reference [43]).

However if we perform the same calculation for the comoving congruence, it follows that
(17)$$\tau \dot{q}=0,$$
where *q* is the heat flux (its radial component) measured by the comoving observer.

Therefore, the departure from equilibrium may occur for the tilted congruence at a time scale smaller than relaxation time, which implies that the Tolman condition is satisfied, whereas for the comoving observer such a departure would occur at a larger time scale, for which the Tolman condition is violated. In other words, the system looks more stable, that is, it takes longer to force its departure from equilibrium, for the comoving observer than for the tilted one. Although we have resorted to the transport Equation (Equation 15), it is a simple matter to see that a similar conclusion is reached by using any other transport equation fulfilling causality and including the “thermal inertial” term.

## 4. Gravitational Radiation and Radiated Information

Radiation (at classical level and for any type of field) may be regarded as the mechanism by means of which the source of the field informs to all observers about changes in its structure and/or changes in its state of motion. This “informational” description of radiation is very well illustrated in the Bondi formalism [44,45]. Thus, when there is a change in the multipole structure of the source, the information about this change must propagate outwardly so that local observers get informed about the new multipole structure of the source. This information is propagated via radiation, therefore when the burst of radiation attains the observers outside the source, such observers absorb the information about the change in the multipole structure. Once the “old” multipole structure has been replaced by the “new” one, one may said that the corresponding information has been erased, in the same sense that we say that reseting an apparatus storing one bit of information amounts to erase one bit of information.

Indeed, in the Bondi approach it is clearly shown that the information required to forecast the evolution of the system, besides the initial data, is contained in a function (one in the axially symmetric case, two in the general case) called “news function” by Bondi, which are identified with gravitational radiation itself. Thus whatever happens at the source and the ensuing changes in the field are related through the news function. Furthermore the total energy of the system is constant if and only if the news function vanishes. Therefore, radiation conveys energy and information. The above comments imply that in the process of radiation information is erased (in the sense above) at the source, which according to Landauer principle entails dissipation of energy. This last conclusion is well known [46] and obeys to the fact that if the source produces gravitational radiation, which is an irreversible process, then an entropy production factor should be present in its hydrodynamic description and therefore this fact should show up in the equation of state of the source. Here we see that it is a simple consequence of the Landauer principle and of the fact that gravitational radiation implies radiation(erasure) of information.

The obvious consequence of the presence of these dissipative processes within the source is the existence of a null fluid outside the source [46]. In other words, the space-time outside the source of gravitational radiation is necessarily non empty, and therefore the implicit assumption in the Bondi formalism considering a vacuum space-time outside the source must be regarded as an approximation. All this having been said, two questions arise:What part of radiated energy corresponds to radiated information?How much information is erased in the process of gravitational radiation?

In what follows, we shall try to answer to these questions.

For this purpose let us consider axially (and reflection) symmetric space-times. For such systems the line element may be written in “Weyl spherical coordinates” as:(18)$$ds^{2}=-A^{2}dt^{2}+B^{2}\left(dr^{2}+r^{2}d\theta^{2}\right)+C^{2}d\phi^{2}+2Gd\theta dt,$$
where A,B,C,G are positive functions of *t*, *r* and θ. We number the coordinates $x^{0}=t,x^{1}=r,x^{2}=\theta ,x^{3}=\phi$. The above line element is valid inside and outside the source.

Now, for an observer at rest in the frame of (Equation 18), the four-velocity vector has components (see Reference [47] for details)
(19)$$V^{\alpha }=\left(\frac{1}{A},0,0,0\right);\;V_{\alpha }=\left(-A,0,\frac{G}{A},0\right),$$
whereas the vorticity vector of the congruence associated to (Equation 19) is
(20)$$\omega^{\alpha }=\left(0,0,0,\omega^{\phi }\right),$$
with
(21)$$\omega^{\phi }=-\frac{\Omega }{C},$$
where the scalar function Ω is given by
(22)$$\Omega =\frac{G(\frac{G^{\prime }}{G}-\frac{2A^{\prime }}{A})}{2B\sqrt{A^{2}B^{2}r^{2}+G^{2}}}.$$
and prime denotes derivative with respect to *r*. From the above expression and regularity conditions it follows that the vorticity vanishes if and only if G=0.

As it was mentioned before, if the source produces gravitational radiation, we have to assume that outside the source there is a null fluid which due to the symmetry constraints has to be described by an energy momentum tensor of the form
(23)$$T_{\alpha \beta }=\lambda l_{\alpha }l_{\beta }+\epsilon n_{\alpha }n_{\beta },$$
where λ and ϵ are two functions of t,r,θ related to the energy density of the null radiation in either direction l and n, and these two null vectors are given by
(24)$$l^{\alpha }=\left(\frac{1}{A},\frac{1}{B},0,0\right)\;n^{\alpha }=\left(\frac{1}{A},0,-\frac{G+\sqrt{A^{2}B^{2}r^{2}+G^{2}}}{AB^{2}r^{2}},0\right).$$

The total energy density of the null radiation is, as usual, defined by
(25)$$\mu_{rad}^{\left(T\right)}=T_{\alpha \beta }V^{\alpha }V^{\beta }=\lambda +\alpha^{2}\epsilon ,$$
where
(26)$$\alpha \equiv 1+\frac{G}{A^{2}B^{2}r^{2}}\left(G+\sqrt{A^{2}B^{2}r^{2}+G^{2}}\right),$$
and the superscript *T* indicates that it defines the energy-density of the radiation produced by all possible irreversible processes occurring at the source.

However, we are interested exclusively in dissipative processes associated to the erasure of information during the emission of gravitational radiation. In order to evaluate the energy density corresponding to these latter processes, we shall resort to a result obtained in Reference [46], according to which if G=0 then the space-time is either static or spherically symmetric (Vaidya). In both cases, of course, no gravitational radiation is produced. Therefore the energy density corresponding to the dissipative processes related to the production of gravitational radiation should be obtained from (Equation 25) by subtracting the contributions of the G=0 case (vorticity free), that is,
(27)$$\mu_{rad}^{\left(L\right)}=\mu_{rad}^{\left(T\right)}-\mu_{rad}^{\left(T,G=0\right)},$$
where the superscript *L* indicates that the energy-density corresponds exclusively to dissipative processes related to the emission of gravitational radiation.

Therefore the total energy dissipated directly related to gravitational radiation is given by
(28)$$E_{rad}^{\left(L\right)}=\int_{r_{\Sigma }}^{\infty }\int_{0}^{\pi }\int_{0}^{2\pi }\sqrt{\left|g\right|}\mu_{rad}^{\left(L\right)}drd\theta d\phi ,$$
where |g| is the absolute value of the determinant of the metric tensor, and the equation of the boundary surface of the source, for simplicity, is assumed to be $r=r_{\Sigma }$.

Next, according to the Landauer principle (Equation 4), the total number *N* of bits erased (radiated) in the process of the emission of gravitational radiation is given by
(29)$$N=\frac{E_{rad}^{\left(L\right)}}{kT\sqrt{|g_{tt}|}\ln2}.$$

On the other hand, as recognized by Bondi himself in Reference [44], the central result of his paper is the expression relating the rate at which the energy is being radiated (by gravitational radiation), with the news function, (his Equation (58)), which reads:(30)$$\frac{dm(u)}{du}=-\frac{1}{2}\int_{0}^{\pi }\frac{dc^{2}\left(u,\theta \right)}{du}\sin\theta d\theta ,$$
where $\frac{dc(u,\theta )}{du}$ is the news function, *u* is the timelike coordinate in the Bondi frame, c(u,θ) is a function entering into the power series expressions of the Bondi metric, and m(u) denotes the energy of the system (the Bondi mass).

Therefore the total radiated energy in the timelike interval $u_{1}\leq u\leq u_{2}$ is given by
(31)$$E_{rad}^{\left(L\right)}=-\int_{u_{1}}^{u_{2}}\left[\frac{1}{2}\int_{0}^{\pi }\frac{dc^{2}\left(u,\theta \right)}{du}\sin\theta d\theta \right]du.$$

Feeding back (Equation 31) into (Equation 29) we find an explicit relationship linking the news function with the total number of bits radiated in the assumed time interval,
(32)$$N=\frac{-\int_{u_{1}}^{u_{2}}\left[\frac{1}{2}\int_{0}^{\pi }\frac{dc^{2}\left(u,\theta \right)}{du}\sin\theta d\theta \right]du.}{kT\sqrt{|g_{tt}|}\ln2}=\frac{\left[\int_{r_{\Sigma }}^{\infty }\int_{0}^{\pi }\int_{0}^{2\pi }\sqrt{\left|g\right|}\mu_{rad}^{\left(L\right)}drd\theta d\phi \right]}{kT\sqrt{|g_{tt}|}\ln2}.$$

Unlike the spherically symmetric case where there is a unique null fluid solution (Vaidya), in the axially symmetric case we have an infinite number of possible solutions. Therefore any explicit expression for the energy density of the null radiation, as well as for the total number of bits erased by emission of gravitational radiation, would depend on the specific solution under consideration.

## 5. Discussion

We have seen so far that resorting to Landauer principle allows us to approach fundamental issues concerning the theory of gravitation from a new perspective, which in turn open the possibility to elucidate quandaries which otherwise would remain without satisfactory explanation. Furthermore, the Landauer principle entails specific physical consequences that could help to discriminate among several scenarios, in the discussion of different self-gravitating systems.

Thus we have seen that in the “orthodox” picture of the Schwarzschild black hole, the fact that information has mass prevents the flow of information from inside the black hole in agreement with the Hawking result according to which the quantum radiation produced by such an object is thermal. However, for the alternative picture described in Section 2, the flow of information is allowed along the θ=0 axis, which in turn implies that the radiation emitted from the black hole is not necessarily thermal. Obviously the last word on this issue must be provided by the experimental observation.

We have also been able to explain the deep discrepancies in the description of the source of a given space-time, as provided by different observers. The fact that different congruences of observers store different amounts of information and the Landauer principle, is all we need to elucidate this quandary.

Finally, we have addressed the problem of gravitational radiation from an “informational” point of view, as suggested by the Bondi formalism. First, as a consequence of the fact that gravitational radiation conveys information, and the Landauer principle, we confirm that an entropy production factor must be present in the source. We were able to measure what part of the total radiation emitted by the source corresponds to gravitational radiation, and what is the total number of bits erased during the radiation process. Furthermore an explicit relationship between the number of erased bits and the Bondi news function was established.

We hope that the results exhibited here would convince my colleagues of the huge potential of information theory to address important issues concerning theoretical physics in general, and theory of gravitation in particular.

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
