# Peer review of "Landauer Principle and General Relativity"

_entropy, 2020, doi:10.3390/e22030340_

Round 1

Reviewer 1 Report

The main goal of the present article is to describe Landauer’s principle in the context of General Relativity. The author explores the fact that information has an associated mass in order to draw some conclusions regarding information, black hole physics and dissipation. Also, the dissipation observed for non-moving observers is analyzed, concluding that dissipation may be frame dependent. As a final consideration, the authors conclude that the emission of gravitational radiation is associated with dissipation as a consequence of Landauer’s principle.

The idea in this paper is intriguing. However, I have some concerns about the main claims of the paper that do not allow me to recommend the paper for publication. My points are the following

1) Eq. (2) was derived from the Einstein mass-energy equivalence along with Landauer principle. The conclusion is that this amount of mass should be assigned to every bit of information stored into a physical system. This sounds very odd. If information has mass, it would be impossible to store information into light (or gravitational waves) since a photon has no rest mass. So, Eq. (2) is valid only for massive particles. But then we get into another problem. We apparently have distinct forms of information, one that has mass and another one that does not have. Note that we simply cannot ascribe a finite mass to a single photon since this would break the entire relativity by attributing a rest reference frame to the photon, whose dispersion relation is not E = mc^2. The author should make this point absolutely clear, since it is fundamental to understand the conclusion that information has mass.

2) As long as the field is weak, I am inclined to accept Eq. (3). However, as equilibrium is a very complicated issue in the presence of gravity (for instance, the physical temperature will not be the same in an extended body as stated by the author), some extra words must be said here. Landauer’s principle is actually an inequality, while Eq (3) is an equality. I am concerned if there is a physical process associated with erasure of information that actually saturates Landauer bound in the presence of gravity. Could the authors comment on this?

3) Eq. (4) is not convincing. How can we actually derive such equation? Under what conditions do we have the equality? How erasure can actually be realized in an arbitrary gravitational field?

4) Bellow Eq. (7), after describing the black hole singularity, the authors conclude that, if information has mass, then no information can come out from the black hole. Well, this happens independently of the mass. A photon, which has no mass, cannot come out as well. The second conclusion, about the fact that the system must be in one single state, I did not understand. Could the author clarify this point? Also, how to properly link this fact with Hawking radiation? A pure state has no information (its entropy is zero), but it is certainly not thermal.

5) Does the discussion bellow Eq. (9) assume that information can travel faster than light? Or am I missing something? That is the only way information can cross the horizon from inside the black hole.

6) The last paragraph of this section seems disconnected from the rest of the discussion. The connection between information, black hole physics and Landauer principle should be precisely stated. In special, how far can we take such connection?

7) Section 3 concludes that while comoving observers observes no dissipation, a different one will do. This conclusion is based on the fact that the latter has more information than the comoving one. It that correct? If yes, the author should clearly explain why a unitary transformation (Lorentz) does not preserve information? This concern also applies to the interpretation in the context of Landauer principle that follows.

8) I completely missed the point of Section 4. What should we conclude from this section? What is the erasure process? How information is erased? If we consider the source and the waves as a closed system, information should not be erased and dissipation should vanish. How can we see this from the calculations of this section?

In summary, I am not convinced by the author’s main claims and, therefore, I cannot recommend the present form of the manuscript for publication.

Author Response

First the reply to referee 1 in the attached covert letter.

Reviewer 2 Report

Referee report is attached.

Author Response

See the reply to referee 2 in the attached covert letter.

Reviewer 3 Report

The article addresses very interesting questions about the relation of the information theory and description of physical processes. Several known examples of such relations including the solution of the Maxwell’s demon paradox are discussed on the base of the Landauer principle. This part of the paper serves as an excellent mini-review. Then the author considers information flows in the processes of gravitation wave emission. An explicit expression which relates the number of bits erased in such a process to the corresponding gravitational functions. The Bondi formalism is applied. An important statement about the possibility to have non-thermal radiation and information flow from a black hole along \theta=0 axis in the is done. Unfortunately, the author did not present for this case concrete predictions which can be verified in observations. But that can be the next step in this line of research.

The paper is written very accurately and clearly. It contains new scientific results and insights for discussion and future work. Certainly, the paper will be interesting for the journal readers.

I recommend to publish the article as it is.

Author Response

The the reply to referee 3 in the attached cover letter.

Round 2

Reviewer 1 Report

The author made several changes in the manuscript, but he did not addressed my main points.

The mass of a particle is a relativistic invariant quantity since it is given by -p^mu p_mu, the contraction of the momentum four vector. Therefore, we cannot ascribe any mass to the photon (or to any massless field). A photon has momentum and energy, but not mass. These things are different. Moreover, as stated in the reply of point 4, the behavior of the photon when passing by the Sun is not because it has mass. The photon simply follows the null geodesic.

The results in the paper are based on the fact that Eq. (1) is the minimum amount of energy that should be associated with a bit of information, based on Landauer’s principle. A few comments are still in order here. Information is basis dependent (observer dependent). Codification of information is a reversible process, in thermodynamic sense. What is irreversible is the erasure of information, which has nothing to do with codification. That is the meaning of Landauer’s principle.

In conclusion, I see no physical or mathematical reason to associate mass to a bit of information.

Moreover, the extension of Landauer’s principle to the relativistic scenario was done by simply changing the temperature by the Tolman one. However, we know that Landauer relation can be actually derived from information theory and its links with thermodynamics (see, for instance Reeb and Wolf for a recent derivation). And, in order to derive this result, interactions and transformations should be defined, including a heat reservoir at the same temperature. This does not happens when gravity comes into play. The authors should describe the information erasure process in oder to convince that Eq. (3) is actually correct.

The discussion regarding the black hole I still did not get. If it is a black hole in our universe (whose metric signature is 2), like the Schwarzchild, then there is no way information can escape, it does not matter the value of the angle. Therefore, I am missing the physical motivation behind this discussion. What kind of black holes are you actually discussing?

In the last but one point the author wrote “…while it is true that comoving and tilted congruences are related by a Lorentz boost, …”. If they are related by a Lorentz transformation, as stated by the authors, then information must be preserved since this is a unitary transformation.

Based on the above comments, I cannot recommend the present manuscript for publication.

Author Response

See the reply on the attached cover letter.

Reviewer 2 Report

All significant corrections were implemented. I have no additional comments.

Author Response

See the attached cover letter.